# MGDC-UNet: Multi-group Deformable Convolution for Medical Image Segmentation

## Abstract

Recently, there has been growing interest in developing Vision Transformer (ViT) or Convolutional Neural Network (CNN) methods for 3D medical image segmentation, which necessitates both large receptive fields and adaptations to varying spatial geometries. Previous works in both CNNs and ViTs demonstrated limitations in capturing the complex spatial and semantic structure of 3D medical images. In this paper, we introduce MGDC-UNet, a multi-group deformable convolution network for 3D volumetric medical image segmentation. Our MGDC-UNet employs deformable convolution operators with learnable spatial offsets to improve attention on semantically important regions. Our approach leverages stable spatial distribution across subjects to enhance semantic learning. We also incorporate transformer components to augment feature learning and reduce inductive biases inherent in traditional CNNs. MGDC-UNet demonstrated superior performance accuracy on three challenging segmentation tasks using public datasets: 1). brain tumor segmentation (BraTS21), 2). CT multi-organ segmentation (FLARE21) and 3). cross-modality MR/CT segmentation (AMOS22). Our network also compared favorably with existing methods in terms of computational efficiency.

## 1 Introduction

Volumetric medical image segmentation plays an important role in the identification and delineation of specific regions, such as tumors or organs, within 3D medical images. In diagnostic and therapeutic applications, this technique aids clinicians in precisely determining the location and scale of pathological changes, which consequently enhances treatment planning and improves patients' quality of life. However, the task of volumetric medical image segmentation is challenging. The complexity of anatomical structures, such as the congestion or even the invasion among tissues, organs, and systems in the limited human body space, may complicate the segmentation process. Additionally, the large volume of 3D image data often demands substantial resources and efficiency.

Previous learning-based approaches have shown remarkable performance in medical image analysis tasks, particularly the U-Net architecture in volumetric image segmentation (Ronneberger et al., 2015). However, existing methods demonstrate limitations in effective receptive fields (ERFs) when dealing with the complicated structure and semantics of volumetric medical image segmentation. Our analysis, illustrated in Fig. 1, shows the ERF distributions of previous network designs lack specificity towards pertinent anatomical structures. Conventional CNN is constrained by its uniform convolution strategy. Since plain convolution kernel samples evenly on the feature map, it underperforms in regions requiring more attention and overcompensates in regions requiring less focus. Small kernel CNN ($3 \times 3 \times 3$) is hindered by a constrained ERF thus offering limited attention and fine-grained analysis capabilities (Fig 1.a). Large kernel CNN ($7 \times 7 \times 7$) improves ERF and local segmentation accuracy but still lacks long-range dependencies (Fig 1.b). Vision Transformers have better attention mechanisms than CNNs, but fall short in capturing semantic correlations due to simple feature correlation design and the complexity of the input volumetric image structures (Fig 1.c). Furthermore, self-attention in ViTs might not inherently focus on the most semantically relevant features of the images, thereby increasing the risk of overfitting.

To address this issue, we propose a novel 3D volumetric feature extraction network designed to explicitly attract more attention to regions with relevant semantics. We observe that, despite their

complexity, medical images often possess strong location-semantics correlations. That is, the position distribution of each organ tends to remain consistent across different subjects. Inspired by Deformable Neural Networks (Zhu et al., 2019b; Wang et al., 2023), we have developed a deformable convolution approach used for 3D volumetric images. Through convolution kernels with learnable position distribution, our network can gather more attention to semantically important regions. Due to the strong correlation between semantics and spatial distribution in 3D volumetric medical images, the learned positional information tends to be more stable, leading to a more efficient and robust network that extracts more semantically accurate features. Our result in Fig. 1d shows a noticeable semantic-related spatial distribution in feature attention.

Specifically, our 3D volumetric medical image segmentation network is named MGDC-UNet. First, we design a learnable spatial offset for each deformable convolution operator which can be applied to 3D volumetric data. The network can adaptively adjust the offset of sampled locations, concentrating its attention on semantically relevant organ positions. This design leverages the stable positional prior of organs to capture robust semantic features. Furthermore, we dynamically adjust offsets and modulation scalars to mitigate the inductive biases inherent in traditional CNNs, achieving transformer-like spatial aggregation. Finally, we designed MGDC blocks with a hybrid deformable convolution and multi-layer perceptron (MLP) structure for effective channel scaling and enhanced feature learning.

Our contributions are as follows:

- We proposed a core operator named MGDC, which capitalizes on the correlation between location and semantics in medical images. Our operator achieves more accurate semantic learning through adaptive attentional positional offsets.

- We augment our core MGDC operator with transformer components in the MGDC block to boost feature learning and attain optimal performance.

- We evaluate our proposed architecture on three large publicly available datasets, demonstrating superior performance in terms of segmentation metrics, inference time, and model parameters.

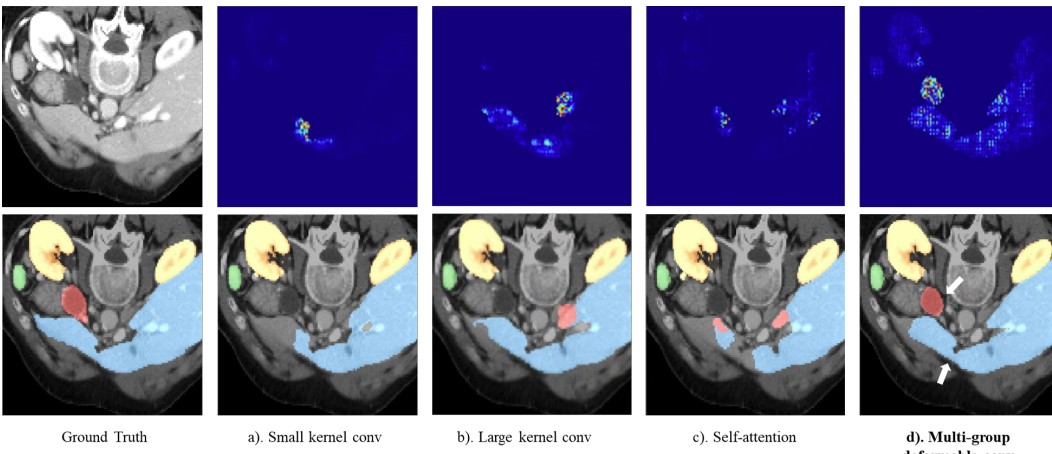

| Ground Truth | a). Small kernel conv | b). Large kernel conv | c). Self-attention | **d). Multi-group deformable conv** |

Figure 1: We compare the ERFs on segmented regions from different operations and their effects on multi-organ segmentation. Top row: ERFs from the bottleneck layer of every method. Bottom row: the segmentation results on an example CT (white arrow indicating improvements). (a) Small-kernel convolutions often segment regions without accounting for the anatomical correlation with adjacent structures. (b) Large-kernel convolutions enhance anatomical context but remain confined to considering only nearby structures. (c) Global self-attention extends the ERF but still falls short in capturing semantic relationships among correlated organs. (d) Multi-group deformable convolutions successfully expand the ERF while adapting to task-specific geometry through learnable offsets, thereby focusing on semantically relevant regions.

## 2 RELATED WORKS

### 2.1 3D VOLUMETRIC MEDICAL IMAGE SEGMENTATION

Due to the strong local inductive bias and parameter efficiency, CNN-based methods have long dominated medical image segmentation. The spatial parameter sharing of CNN enables compact designs suitable for medical image analysis. (Ronneberger et al., 2015) introduced U-Net, a CNN architecture with symmetric expansive and contractive paths enabling precise localization, making it a standard choice for many segmentation tasks in medical imaging. However, the limited receptive field of CNNs can significantly hinder their performance on medical segmentation tasks, where the objects are often irregular or distorted. A standard convolutional layer with a small kernel size can only capture local spatial patterns. Even with pooling or striding, the inherent design of CNNs forces them to accumulate global context through many layers, potentially losing or diluting important long-range information. To address the locality of CNN, variants of U-Net have been proposed by leveraging novel breakthroughs from various vision tasks. Attention UNet utilized attention-gates to select important features to improve segmentation performance (Oktay et al., 2018). (Zhang et al., 2017) introduced dilated convolution and pyramid pooling to U-Net to enlarge the receptive field. Self-attention has also been applied to address the locality of convolution operation (Sinha & Dolz, 2020). Different from previous works on this task, our proposed method can leverage the correlation of spatial prior and semantics in 3D volumetric image segmentation tasks, which gives better attention to semantic relevant regions.

### 2.2 DEFORMABLE CONVOLUTION NEURAL NETWORKS

Deformable convolution has emerged as a powerful technique for addressing the limitations of traditional CNNs in tasks requiring adaptive receptive fields, such as image segmentation. Initial contributions, such as Deformable ConvNet by (Dai et al., 2017) laid the foundation by introducing dynamic offsets to adapt receptive fields. Subsequent advancements, such as DCNv2 and DCNv3, incorporated learnable modulation scalars and multi-group spatial aggregation for greater flexibility and efficiency (Zhu et al., 2019b; Wang et al., 2023). While deformable convolution has been effectively applied in 2D medical image segmentation and 3D CT multi-organ segmentation tasks (Jin et al., 2019; Heinrich et al., 2019), its full potential in combination with transformer-like architectures for 3D medical image segmentation remains underexplored. Our hypothesis is that deformable convolution can significantly augment transformer-like architectures, offering benefits in handling long-range dependencies and providing computational efficiency compared to traditional CNNs and Vision Transformers. Different from previous works, we further equipped deformable convolution with multi-group spatial aggregation and transformer-like components for 3D medical image segmentation, while still improving computational efficiency.

## 3 METHOD

In this section, we first present our MGDC module and block design. To design a large-scale deformable CNN for medical image segmentation, we start by improving the original deformable convolution with multi-group mechanisms to improve feature encoding capabilities. We then design the basic block of MGDC by incorporating transformer components to stronger modeling capacity.

### 3.1 MULTI-GROUP DEFORMABLE CONVOLUTION

While traditional CNNs typically use small convolution kernels that result in limited effective receptive fields, deformable convolution enhances the conventional convolutional process by allowing for adaptive sampling positions within the convolutional grid. Unlike traditional convolution, which operates on uniformly spaced grid points, deformable convolution modifies these positions based on learnable offsets, thus enabling the model to learn more flexible representations of the input. Accordingly, we first take a 3D a dynamic deformable convolution network (3D DCN) (Zhu et al., 2019b) with adaptive sampling offsets and modulation masks to enhance the targeted segmentation tasks. Given an input $x \in R^{H \times W \times D}$ and a current voxel $v_0$, our proposed 3D DCN layer can be

formulated in the following:

$$y(v_0) = \sum_{s=1}^{S} w_s m_s x(v_0 + v_s + \Delta_{v_s}) \tag{1}$$

where $s$ enumerates the sampling points with a total of $S$ points. $v_s$ represents the $s$-th location of the pre-defined grid sampling $\{(-1,-1,-1), (-1,-1,0), \ldots, (1,1,0), (1,1,1)\}$ as in regular $3 \times 3 \times 3$ convolutions. $\Delta v_s$ is the offset corresponding to the $s$-th sampling location, $w_s$ denotes the projection weights of the $s$-th sampling point, and $m_s$ is the modulation scalar of the $s$-th sampling point normalized by sigmoid function. From equation (1), we can see that the sampling offset $\Delta v_s$ is conditioned based on inputs and is able to achieve both short and long-range dependencies. Furthermore, the modulation scalar $m_s$ is also learnable and dynamically adjusted based on inputs. Therefore, the 3D DCN layer already shares similar properties with MHSA. Nonetheless, the proposed 3D DCN layer faces challenges in medical image segmentation. First, the design leads to linear memory complexity and computational demands, raising the risk of overfitting in data-limited medical settings. Second, unlike transformers or group convolutions, DCN lack a multi-group mechanism to capture diverse features, limiting their representational power.

To address these limitations, we introduce MGDC, a specialized deformable convolution operator. To remedy the computation complexity, we propose to use depth-wise convolution and detach the regular convolution $w_s$ into depth-wise and point-wise parts. The depth-wise part is responsible for the location-aware modulation scalar $m_k$ and the point-wise part is the shared projection weights $w_g$ among sampling points. We also introduce multi-group spatial aggregation to effectively learn richer information from different representation subspaces at different locations. Similar to the concept of grouped convolution, we split the spatial aggregation process into $G$ groups, each of which has individual sampling offsets $\Delta p_{gs}$ and modulation scaler $m_{gs}$ and hence different groups on a single convolution layer can have different spatial aggregation patterns, resulting in stronger features for downstream tasks. $m_{gs}$ and $\Delta v_{gs}$ are obtained via two linear layers applied over input. Given an input $x$, our proposed MGDC can be formulated as the following:

$$x_1 = DWC(x) \tag{2}$$

$$\Delta v_{gs} = \text{linear}(x_1) \tag{3}$$

$$m_{gs} = \text{softmax}(\text{linear}(x_1), S) \tag{4}$$

$$y(v_0) = \sum_{g=1}^{G} \sum_{s=1}^{S} w_g m_{gs} x(v_0 + v_s + \Delta v_{gs}) \tag{5}$$

where DWC stands for depth-wise convolution and linear stands for linear transformation. $S$ stands for the total number of sampled points. $G$ denotes the total number of aggregation groups. For the $g$-th group, $w_g$ denotes the location-irrelevant projection weights of the group, $w_g \in \mathbb{R}^{C_g \times C_g}$ where $C_g = C/G$ represents the group channel dimension. $m_{gs}$ denotes the modulation scalar of the $s$-th sampling point in the $g$-th group, normalized by the softmax function along dimension $S$. $x_g \in \mathbb{R}^{C_g \times H \times W \times D}$ represents the $g$-th grouped input feature map. $\Delta v_{gs}$ is the offset corresponding to the grid sampling location $p_s$ in the $g$-th group. Since $v_0 + v_s + \Delta v_{gs}$ might be fractional, trilinear interpolation is used to convert fractions to integers.

## 3.2 MGDC-UNET

The overall pipeline of our proposed method is illustrated in Figure 2. Following the encoder-decoder design of Hatamizadeh et al. (2022), our MGDC-UNet consists of four stages in encoder, decoder, and four residual connections. For an input volume with a size of $H \times W \times D$, MGDC-UNet first leverages two convolution embedding layers to obtain downsampled feature maps of $\frac{H}{4} \times \frac{W}{4} \times \frac{D}{4} \times C$, where we set $C$ empirically to 48. Next, each stage of encoding starts with MGDC blocks to extract spatial representations and ends with a downsample block (except for the last stage) to produce hierarchical features and double the channel dimension. After hierarchical encoding, the output from each stage in the encoder is fed to a CNN-based decoder with skip connections. Inside the decoder, a transposed convolutional layer is used for upsampling input and concatenating with

multi-scale features. On the final layer, we concatenated the transformed input with the upsampled features to produce the final segmentation map Below we show detailed design of the MGDC blocks.

**1). MGDC Block:** We present the MGDC Block, a new architecture that includes a reverse bottleneck design similar to MobileNetV2 (Sandler et al., 2018), but augmented with transformer components. While traditional inverted bottleneck design utilized depthwise convolution, our MGDC block leverages two MLP layers for channel expansion and reduction and LayerNorm for normalization, a design further inspired by Vision Transformers. This approach enables the network to capture more complex and richer features. Given input into the MLP layer $m_{\text{in}}$, we define MLP function as:

$$\text{MLP} = \text{LN}\left(\text{Linear}\left(\text{GELU}\left(\text{Linear}\left(m_{\text{in}}\right)\right)\right)\right) \tag{6}$$

The overall block is formulated by the MLP layer with the GELU activation and post-normalization strategy as:

$$x' = x + \text{GELU}\left(\text{LN}\left(\text{MGDC}\left(x\right)\right)\right) \tag{7}$$

$$x_{\text{out}} = x' + \text{LN}\left(\text{MLP}\left(x'\right)\right) \tag{8}$$

**2). Stem block & downsample block:** Hierarchical design downsamples the input to varying resolutions to extract multi-scale features and is commonly used in image segmentation. To obtain hierarchical feature maps, our stem block first reduces the input resolution by a factor of 4. We stack two plain convolution layers with a stride of 2, two Layer Normalization layers, and one GELU activation layer. The downsample block only reduces the input feature by a factor of 2. It consists of one plain convolution with a stride of 2, followed by one Layer Normalization layer.

**3). Upsample block & final block:** To upsample the processed feature maps, we utilize transposed convolution with a stride of 2 (except for the last upsample block which uses a stride of 4), followed by Instance Normalization. An additional plain convolution layer is used to further extract semantic information from the decoded feature maps. In the final block, we swap the transposed convolution with plain convolution and output the segmentation maps.

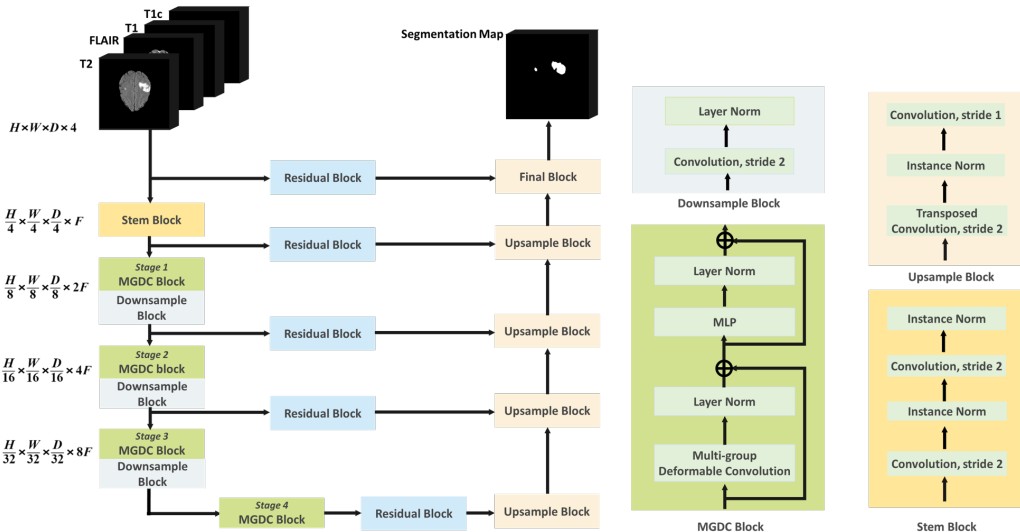

Figure 2: Illustration of Proposed MGDC-UNet Architecture. The complete encoder-decoder architecture is displayed on the left. Structures of MGDC block, stem block, downsample block and upsample block are revealed on the right.

## 4 RESULTS

### 4.1 IMPLEMENTATION DETAILS AND DATASET

To evaluate the proposed MGDC-UNet, we trained and evaluated the network in BraTS21, FLARE 2021, and AMOS 2022 dataset on an NVIDIA A6000. A comprehensive overview of datasets and

evaluation strategies can be found in Appendix A.1. The BraTS21 dataset for glioma segmentation includes 1,251 multi-parametric MRI scans with four modalities and evaluates using Dice score (DSC) and 95% Hausdorff distance (HD95). Annotations target three sub-regions: Gd-enhancing tumor (ET), peritumoral tissue (ED), and necrotic core (NCR). FLARE 2021 dataset for abdominal organ segmentation consists of 361 multi-contrast CT scans from two major medical centers and involves verification from five radiologists. For AMOS 2022, we focused on cross-modality CT-MRI segmentation using 300 CT and 60 MRI scans. Annotations were performed for 15 abdominal organs by multiple groups of radiologists. Both the Dice score and surface Dice score were computed for FLARE 2021 and AMOS 2022. A comprehensive overview of our training procedure can be found Appendix A.2. In all experiments, the networks are optimized by the AdamW optimizer with a linear warmup and cosine annealing strategy. For the BraTS21 dataset, we opted for an input size of $(128, 128, 128)$ following the methodology established by (Wang et al., 2021). On the other hand, for the AMOS and FLARE datasets, an input size of $(96, 96, 96)$ was employed, as suggested by Lee et al. (2022). Several techniques including random rotation, random flipping, random cropping, random intensity shifts, and random affine transformations were deployed. Additionally, to fully demonstrate the capability of the DCN layer in handling large kernels for performance enhancement, we conducted experiments using various convolution kernel sizes (3, 5, and 7) to maximize MGDC's performance.

## 4.2 COMPARISONS WITH STATE-OF-THE-ART METHODS

To demonstrate the effectiveness of our proposed method, we compare it against state-of-the-art CNNs, transformers, and ConvNext methods on volumetric segmentation tasks. Our comparative methods include ResUNET (Zhang et al., 2018), SegResNet (Myronenko, 2019), Swin UNETR (Tang et al., 2022), TransBTS (Wang et al., 2021) and UXNET (Lee et al., 2022). We reimplemented the above methods according to the publicly released codes. To ensure the fairness of the comparison, we utilized the same optimization tool, data augmentation strategies, and data split for each method. We conducted five-fold cross-validation on each dataset respectively, and paired student's t-test was used to evaluate statistical significance.

**1). Experiment results on BRaTS21 dataset:** Table 1 presents a comparative analysis of MGDC-UNet with state-of-the-art segmentation techniques on the BraTS21 dataset. Notably, MGDC-UNet outperformed all competing methods, registering remarkable improvements in both the DSC and HD95. For a kernel size of 3, the MGDC-UNet achieved a DSC score of 90.6% and an HD95 value of 4.816 mm, surpassing Swin UNETR by 0.9% and 0.849 mm, respectively. Further investigation revealed consistent performance gains when incrementing the kernel size from 3 to 5 and ultimately to 7, corroborating our theory that larger receptive fields improve segmentation performance. A paired t-test provided additional statistical validation for the observed enhancements when increasing the kernel size from 3 to 7. For a deeper visual understanding, we refer the reader to Figure 3. As depicted in the first and second rows, our MGDC-UNet effectively minimizes false positive NCR (red) and ET (yellow) regions when segmenting brain tumors compared to competing methods. The third row also clearly illustrates MGDC-UNet's exceptional accuracy in outlining various tumor boundaries. Our observations further revealed that even with a small kernel size ($k = 3$), our MGDC-UNet still excelled over the large-kernel ConvNext method, UXNET, by 0.9% in DSC. This indicates that deformable convolutions are capable of capturing long-range dependencies efficiently. Statistical validation reinforced the superior performance of MGDC-UNet over the best SOTA methods.

Furthermore, we provide the time efficiency and the memory usage of MGDC-UNet and comparison methods. For CNN methods, although ResUNet and SegResNet demonstrated fast training and inference time, their segmentation performances were much worse than our MGDC-UNet. For transformer methods, Swin UNETR outperformed TransBTS in segmentation accuracy but demonstrated lower training and inference speed. Compared to MGDC-UNet, both methods still have relatively high memory consumption. For the ConvNext method, UXNET demonstrated a good balance between performance and training speed. However, our MGDC-UNet $k = 3$ is 38% faster and has 19% less memory consumption than UXNET while still improving DSC by 1.3%. Therefore, our model achieves the best balance between segmentation performance and time-resource efficiency.

**2). Experiment results on FLARE21 dataset:** As shown in Table 2, our MGDC-UNet outperformed all comparable methods in terms of DSC and SDC. Notably, MGDC-UNet outperformed

Table 1: Quantitative comparison with SOTA methods in BraTS21 dataset with Avg (average) results. The best result from SOTA methods is underlined. T-test is performed between the best result from SOTA models and our models. **Bold** means p-value $p < 0.05$. Efficiency analysis was also performed in terms of time (training or inference on each sample) and memory consumption for various models.

| Methods | DSC | | | | HD95 (mm) | | | | Time (s) | | Memory (G) |
|---------|-----|-----|-----|-----|-----------|-----|-----|-----|----------|-----------|------------|
| | TC | WT | ET | Avg | TC | WT | ET | Avg | Train | Inference | |
| ResUNET | 0.875 | 0.912 | 0.858 | 0.881 | 7.740 | 12.446 | 6.542 | 8.912 | 0.25 | 0.37 | 2.5 |
| SegResNet | 0.901 | 0.917 | 0.867 | 0.895 | 6.481 | 10.421 | 5.478 | 7.460 | 0.24 | 0.78 | 3.3 |
| UXNET | 0.890 | 0.916 | 0.873 | 0.893 | 7.442 | 9.583 | 5.053 | 7.357 | 0.63 | 2.7 | 10.3 |
| Swin UNETR | 0.898 | 0.921 | 0.872 | 0.897 | 5.091 | 7.770 | 4.135 | 5.665 | 0.55 | 2.56 | 11.4 |
| TransBTS | 0.864 | 0.907 | 0.838 | 0.869 | 8.651 | 10.972 | 7.385 | 9.003 | 0.36 | 1.69 | 9.6 |
| MGDC-UNet (k=3) | 0.908 | 0.928 | **0.881** | **0.906** | 4.774 | **6.024** | 3.951 | **4.816** | 0.39 | 1.67 | 8.3 |
| MGDC-UNet (k=5) | **0.911** | **0.933** | **0.885** | **0.910** | **4.083** | **5.880** | 3.787 | **4.583** | 0.45 | 1.89 | 8.6 |
| MGDC-UNet (k=7) | **0.917** | **0.936** | **0.888** | **0.914** | **3.818** | **5.504** | 3.605 | **4.309** | 0.51 | 2.08 | 9.4 |

a). T2    b). T1    c). T1CE    d). FLAIR    e). Ground Truth    f). ResUNet    g). SegResNet    h). Swin UNETR    i). TransBTS    j). UXNET    **k). MGDC-UNet (ours)**

Figure 3: Visualization of segmentation results on BraTS21 dataset. Green, yellow and red regions indicate ED, ET and NCR.

UXNET (the previous state-of-the-art on Flare 21) by $0.8\%$ in DSC and $0.4\%$ in SDC. Experiments on enlarging the kernel size showed that MGDC-UNet achieved the best performance when $k = 7$, achieving $94.4\%$ DSC and $94.1\%$ SDC. We also generated visualization results in Figure 4. MGDC-UNet demonstrated the best segmentation performance for kidneys (row 2) and reduced false negative regions for liver segmentation (row 3).

**3). Experiment results on AMOS22 dataset:** Table 3 summarizes results on the AMOS 22 dataset. Our MGDC-UNet (k=3) outperformed all comparable methods on CT segmentation tasks in both DSC and SDC. For MRI segmentation, both SegResNet and UXNET demonstrated similar performance to MGDC-UNet (k=3) in terms of DSC. However, after switching kernel size to 7, MGDC-UNET outperformed both methods by $0.3\%$ in DSC. For SDC, all MGDC-UNet models demonstrated superior performance, leading comparison methods by $0.7\%$ to $4.6\%$. While cross-modal multi-organ segmentation still remained a challenge, our MGDC-UNet still achieved satisfactory performance for most organs (Figure 5, row two). In row one, we found that MGDC-UNet provided finer segmentation details of the stomach than other methods.

## 4.3 ABLATION STUDY

**1). Effectiveness of MGDC operator:** We started by investigating the effectiveness of our proposed MGDC operator. As shown in Table 4 (row 1 and 2), introducing a shared weight mechanism to MGDC decreased $22\%$ parameters. Our MGDC introduced shared weights to alleviate the high computational costs and reduce memory consumption by $33\%$. We also observed a small performance boost after switching from 3D DCN to MGDC. Next, we compared the MGDC with and

Table 2: Quantitative comparison with SOTA methods in FLARE21 dataset with Avg (average) results. The best result from SOTA methods is underlined. T-test is performed between the best result from SOTA models and our models. **Bold** means p-value $p < 0.05$.

| Methods | DSC | | | | | SDC | | | | |
|---|---|---|---|---|---|---|---|---|---|---|
| | Spleen | Kidney | Liver | Pancreas | Avg | Spleen | Kidney | Liver | Pancreas | Avg |
| ResUNET | 0.976 | 0.955 | 0.968 | 0.774 | 0.918 | 0.957 | 0.958 | 0.986 | 0.726 | 0.907 |
| SegResNet | 0.976 | 0.956 | 0.969 | 0.816 | 0.929 | 0.966 | 0.965 | 0.992 | 0.799 | 0.930 |
| UXNET | 0.977 | 0.959 | 0.973 | 0.819 | 0.932 | 0.966 | 0.967 | 0.994 | 0.810 | 0.934 |
| Swin UNETR | 0.978 | 0.959 | 0.971 | 0.803 | 0.928 | 0.965 | 0.963 | 0.986 | 0.782 | 0.924 |
| TransBTS | 0.978 | 0.959 | 0.971 | 0.764 | 0.918 | 0.968 | 0.966 | 0.991 | 0.719 | 0.911 |
| MGDC-UNet (k=3) | 0.982 | 0.963 | 0.972 | **0.842** | **0.940** | 0.973 | 0.967 | 0.992 | **0.819** | 0.938 |
| MGDC-UNet (k=5) | **0.992** | 0.965 | 0.967 | **0.840** | **0.941** | 0.973 | 0.968 | 0.994 | **0.823** | 0.940 |
| MGDC-UNet (k=7) | **0.995** | **0.968** | **0.971** | **0.843** | **0.944** | **0.975** | 0.969 | 0.994 | **0.825** | **0.941** |

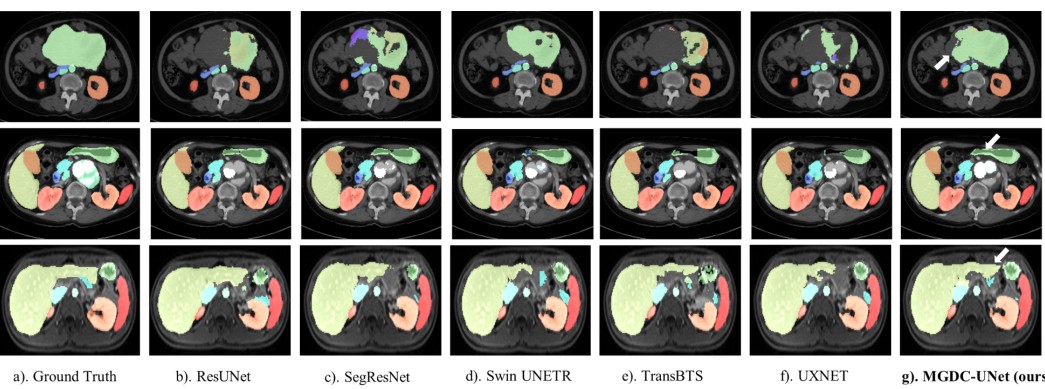

a). Ground Truth  b). ResUNet  c). SegResNet  d). Swin UNETR  e). TransBTS  f). UXNET  **g). MGDC-UNet (ours)**

Figure 4: Visualization of segmentation results on FLARE21 dataset. White arrow indicates superior regions of our results compared with other models

a). Ground Truth  b). ResUNet  c). SegResNet  d). Swin UNETR  e). TransBTS  f). UXNET  **g). MGDC-UNet (ours)**

Figure 5: Visualization of segmentation results on AMOS22 dataset. White arrow indicates superior regions of our results compared with other models

without the multi-group spatial aggregation. As shown in row 2 and row 3, introducing a multi-group mechanism into deformable convolution improved DSC by $0.5\%$ for brain tumor segmentation and $0.4\%$ for multi-organ segmentation. We suspected that a larger training sample size would further improve the performance gains of the larger kernel convolution method. In this section, we study how the different components in our designed MGDC-UNet contribute to gains in segmenta-

Table 3: Quantitative comparison with SOTA methods in AMOS22 dataset with Avg (average) results. The best result from SOTA methods is underlined. T-test is performed between the best result from SOTA models and our models. **Bold** means p-value $p < 0.05$.

| Methods | DSC | | | SDC | | |
|---|---|---|---|---|---|---|
| | CT | MRI | Avg | CT | MRI | Avg |
| ResUNET | 0.825 | 0.706 | 0.805 | 0.840 | 0.823 | 0.846 |
| SegResNet | 0.854 | 0.720 | 0.830 | 0.888 | 0.867 | 0.885 |
| UXNET | 0.856 | 0.720 | 0.833 | 0.886 | 0.860 | 0.882 |
| Swin UNETR | 0.851 | 0.712 | 0.828 | 0.876 | 0.862 | 0.874 |
| TransBTS | 0.847 | 0.717 | 0.826 | 0.877 | 0.858 | 0.873 |
| MGDC-UNet (k=3) | **0.865** | 0.720 | **0.840** | 0.893 | **0.884** | 0.891 |
| MGDC-UNet (k=5) | **0.865** | 0.721 | **0.841** | 0.894 | **0.886** | **0.892** |
| MGDC-UNet (k=7) | **0.866** | 0.723 | **0.841** | 0.894 | **0.885** | **0.892** |

Table 4: Ablation on component of MGDC-UNet. Network parameters and DSC from BraTS21 and FLARE21 were reported.

| Operator | Multi-group | MLP | Params (M) | BraTS21 | FLARE21 |
|---|---|---|---|---|---|
| 3D DCN | ✗ | ✗ | 71.5 | 0.894 | 0.929 |
| MGDC | ✗ | ✗ | 58.7 | 0.896 | 0.930 |
| MGDC | ✓ | ✗ | 58.1 | 0.901 | 0.932 |
| MGDC | ✓ | ✓ | 61.2 | 0.906 | 0.940 |

tion performance. We conducted ablation studies on BraTS21 and Flare21 datasets due to their large sample sizes. All ablation studies on MGDC-UNet were performed with kernel size set to 3.

**2). Effectiveness of MGDC Block:** The core design of our MGDC Block is introducing a multi-layer perceptron as a feed-forward network. As shown in Table 4 (row 3 and 4), introducing MLP layers to the network successfully scaled up the model and further improved segmentation performance by $0.5\%$ and $0.8\%$ in DSC on BraTS21 and FLARE21 datasets. This also confirmed our hypothesis that transformer-like components can also enhance medical image segmentation.

## 5 CONCLUSION AND DISCUSSION

In this paper, we introduce MGDC-UNet, the first 3D multi-group deformable convolution network for medical image segmentation. Our architecture integrates multi-group spatial aggregation into deformable convolutions, inspired by the multi-head mechanism found in ViTs. Additionally, we incorporate transformer-specific elements such as MLP and LayerNorm to emulate the inverted-bottleneck design featured in ViT blocks. To further enhance performance, we explore the use of large deformable convolutional kernels, which further improve the network's capability for capturing long-range dependencies—crucial for achieving high-quality segmentation results. Our rigorous evaluation clearly demonstrates MGDC-UNet's advantages through both quantitative and statistical metrics, establishing its superiority over existing methods. MGDC-UNet excels in capturing long-range dependencies, a feat attributed mainly to its flexible offsets and modulation scalars. This distinctive feature sets our model apart from traditional CNNs, which frequently struggle with global attention, a limitation we overcome as demonstrated in Figure 1. When compared to transformer-based architectures, MGDC-UNet offers dual benefits: it not only learns more robust representations but also achieves this with fewer model parameters. Thus, MGDC-UNet emerges as a resilient solution, less prone to overfitting while maintaining higher computational efficiency. In summary, MGDC-UNet surpasses the state-of-the-art transformer models in performance with less memory usage and better performance speed across three challenging public datasets. We believe that MGDC-UNet holds significant potential as a tool for fast organ delineation in clinical applications.

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

# A APPENDIX

## A.1 DATA STRATEGIES FOR TRAINING AND EVALUATION

Table 5: Data strategies and evaluation details for each public datset.

| Settings | BraTS21 | FLARE21 | AMOS22 | Prostate158 |
|---|---|---|---|---|
| Imaging Modality | MRI: T2W, T1CE, T1, Flair | CT | CT or MRI | MRI: T2W, DWI, ADC |
| Total samples | 1251 | 361 | 360 | 158 |
| Cross validation | 5-fold CV | 5-fold CV | 5-fold CV | 5-fold CV |
| Train/Val/Test | Train 850, Val 150, Test 251 | Train 246, Val 43, Test 72 | Train 204, Val 36, Test 120 | Train 118, Val 21, Test 19 |
| Metrics | Dice, Hausdorff Distance | Dice, Surface Dice | Dice, Surface Dice | Dice, Hausdorff Distance |

## A.2 TRAINING RECIPES FOR ALL DATASETS

Table 6: Training configurations for each dataset. For BraTS21, we followed preprocessing pipeline as (Tang et al., 2022). For FLARE21 and AMOS22, we followed preprocessing pipeline as (Lee et al., 2022). Preprocessing method for Prostate158 was discussed in A.3.

| Settings | BraTS21 | FLARE21 | AMOS22 | Prostate158 |
|---|---|---|---|---|
| Channels | 48, 96, 192, 384 | | | |
| Channel groups | 3, 6, 12, 24 | | | |
| MLP embedding ratio | 4, 4, 4, 4 | | | |
| Depths | 2, 2, 2, 2 | | | |
| Input size | 128, 128, 128 | 96, 96, 96 | 96, 96, 96 | 160, 160, 32 |
| Batch | 1 | 1 | 1 | 4 |
| LR, schedule | 5e-5, cosine | 1e-4, cosine | 1e-4, cosine | 1e-5, cosine |
| Epoch | 300 | 500 | 500 | 100 |
| Warmup epoch | 20 | 20 | 20 | ✕ |
| Loss | Dice | DiceCE | DiceCE | Generalized Dice |
| Random Flip | ✓ | ✕ | ✕ | ✓ |
| Random Crop | ✓ | ✓ | ✓ | ✕ |
| Random Rotation | ✓ | ✕ | ✕ | ✓ |
| Random Intensity Shift | ✓ | ✓ | ✓ | ✕ |
| Random Affine | ✕ | ✓ | ✓ | ✕ |

## A.3 ADDITIONAL EXPERIMENTS ON PROSTATE158

Automated segmentation of prostate MR images is crucial in clinical settings, but it faces significant challenges. Accurately segmenting the prostate from MRI is difficult due to unclear boundaries with adjacent tissues (Zhu et al., 2019a). The complexity of prostate cancer, which varies in size, shape, and texture, poses additional challenges for automated deep learning methods.

We evaluated MGDC-UNet on the Prostate158 dataset (Adams et al., 2022), which includes 158 prostate MRIs with T2w, diffusion-weighted imaging (DWI) sequences and anisotropic diffusion coefficient (ADC) maps, for two tasks: segmenting the prostate gland and clinically significant prostate cancer. The MRIs were annotated by two board-certified radiologists. For preprocessing, we followed (Saha et al., 2021) where images were resampled to a uniform axial resolution and slice thickness, followed by center cropping of the prostate to a standard size and interpolating the final image to fit our network's input requirements.

As shown in Table 7, MGDC-UNet still exhibited superior performance under a smaller dataset size. For prostate region segmentation, small kernel MGDC-UNet (k=3) significantly outperformed all comparable methods, achieving a DSC of 0.855 and HD95 of 5.06. Further increasing kernel size to 5 led to 1.0% percent improvement in DSC and the best performance was achieved at k=7, with 0.866 DSC. For prostate cancer segmentation, the small kernel MGDC-UNet (k=3) outperformed other state-of-the-art methods, achieving a DSC of 0.515 and HD95 of 7.77. Increasing kernel size to 5 and 7 further led to 0.9% and 1.2% improvement in DSC. We believed that larger MGDC kernels capture a broader spatial context, which is beneficial for difficult segmentation tasks involving irregular and heterogeneous objects such as prostate gland and clinically significant prostate cancer.

Table 7: Quantitative comparison with SOTA methods in Prostate158 dataset. The best result from SOTA methods is underlined. T-test is performed between the best result from SOTA models and our models. **Bold** means p-value $p < 0.05$.

| Methods | Prostate | | Cancer | |
|---|---|---|---|---|
| | DSC | HD95 | DSC | HD95 |
| ResUNET | 0.812 | 8.56 | 0.382 | 13.59 |
| SegResNet | 0.833 | 6.26 | 0.497 | 8.64 |
| UXNET | 0.842 | 6.03 | 0.453 | 9.02 |
| Swin UNETR | 0.824 | 6.31 | 0.417 | 10.92 |
| TransBTS | 0.840 | 6.08 | 0.400 | 11.81 |
| MGDC-UNet (k=3) | **0.855** | **5.06** | **0.515** | 7.77 |
| MGDC-UNet (k=5) | **0.865** | **4.87** | **0.524** | **6.89** |
| MGDC-UNet (k=7) | **0.866** | **4.56** | **0.527** | **6.62** |

## A.4 LIMITATIONS

While MGDC-UNet demonstrated excellent performances across several datasets, our work is not without limitations. While we incorporated transformer components such as MLP and Layernorm to enhance feature learning, such components could also benefit from parameter scaling similar to other vision transformer backbones. In our future directions, we aim to explore scaling up MGDC-UNet following EfficientNet's compound scaling technique to balance depth, width and resolution of CNNs. By systematically scaling these dimensions with a set of fixed scaling coefficients, we can potentially build a foundational CNN model for medical image segmentation.

Also, our observations on the Prostate158 dataset align with the notion that kernel size can significantly influence feature learning and hence segmentation accuracy. Larger kernels improves segmentation performance by more effectively enlarging the ERFs, but at the cost of increased computational complexity. To address this, our future work will focus on refining MGDC-UNet through a heterogeneous kernel approach, inspired by the Inception architecture. This strategy will integrate various kernel sizes, enabling efficient multi-scale information processing and yielding more accurate segmentation with optimized computational efficiency.

## A.5 Memory and time efficiency analysis

Table 8: Time (training or inference on each sample) and memory efficiency analysis for BraTS21, FLARE21 and AMOS22. Performance gaps in DSC were also shown with respect to MGDC-UNet (k=3).

| Dataset | Method | Memory (G) | Time (s) | | $\Delta DSC$ |
|---|---|---|---|---|---|
| | | | Train | Inference | |
| BraTS21 | ResUNET | 2.5 | 0.25 | 0.37 | $-2.5\%$ |
| | SegResNet | 3.3 | 0.24 | 0.78 | $-1.1\%$ |
| | UXNET | 10.3 | 0.63 | 2.70 | $-1.3\%$ |
| | Swin UNETR | 11.4 | 0.55 | 2.56 | $-0.9\%$ |
| | TransBTS | 9.6 | 0.36 | 1.69 | $-3.7\%$ |
| | MGDC-UNet (k=3) | 8.3 | 0.39 | 1.67 | $0.0\%$ |
| | MGDC-UNet (k=5) | 8.6 | 0.45 | 1.89 | $+0.4\%$ |
| | MGDC-UNet (k=7) | 9.4 | 0.51 | 2.08 | $+0.4\%$ |
| FLARE21 | ResUNET | 4.0 | 0.80 | 0.96 | $-2.2\%$ |
| | SegResNet | 4.8 | 0.74 | 1.16 | $-1.1\%$ |
| | UXNET | 8.9 | 1.05 | 2.89 | $-0.8\%$ |
| | Swin UNETR | 10.3 | 0.77 | 2.81 | $-1.2\%$ |
| | TransBTS | 7.6 | 0.89 | 1.25 | $-2.2\%$ |
| | MGDC-UNet (k=3) | 7.6 | 0.77 | 1.93 | $0.0\%$ |
| | MGDC-UNet (k=5) | 8.4 | 0.79 | 2.03 | $+0.1\%$ |
| | MGDC-UNet (k=7) | 9.3 | 0.79 | 2.35 | $+0.4\%$ |
| AMOS22 | ResUNET | 7.0 | 1.22 | 1.51 | $-3.5\%$ |
| | SegResNet | 8.4 | 1.12 | 2.09 | $-1.0\%$ |
| | UXNET | 15.0 | 1.40 | 5.90 | $-0.7\%$ |
| | Swin UNETR | 16.7 | 1.20 | 4.91 | $-1.2\%$ |
| | TransBTS | 13.2 | 1.23 | 2.50 | $-1.4\%$ |
| | MGDC-UNet (k=3) | 10.1 | 1.19 | 3.52 | $0.0\%$ |
| | MGDC-UNet (k=5) | 10.4 | 1.23 | 3.60 | $+0.1\%$ |
| | MGDC-UNet (k=7) | 10.6 | 1.16 | 4.18 | $+0.1\%$ |

