# OpenReview forum: "MGDC-UNet: Multi-group Deformable Convolution for Medical Image Segmentation"
_ICLR.cc/2024/Conference — Submitted to ICLR 2024_

### Official Review · Reviewer_sKkf · 2023-11-02

**Soundness:** 2 fair
**Presentation:** 3 good
**Contribution:** 3 good
**Rating:** 8
**Confidence:** 4

**Summary:**

They introduce MGDC-UNet, a multi-group deformable convolution network for 3D volumetric medical image segmentation. Their MGDCUNet employs deformable convolution operators with learnable spatial offsets to improve attention on semantically important regions.


They use three challenging segmentation tasks using public datasets:

1) brain tumor segmentation (BraTS21)

 2)  CT multi-organ segmentation (FLARE21)

 3)  cross-modality MR/CT segmentation (AMOS22)

MGDC-UNet demonstrated superior performance accuracy in the three challenging segmentation tasks.

**Strengths:**

Well organized and clearly written.

The effectiveness of proposed method has been well supported by experiments.

**Weaknesses:**

The did not mention the limitation of their method.

They did not add results of their model for small dataset like MRBrainS dataset and MICCAI iSEG dataset (minor issue).

**Questions:**

could you change your title?  Because (This paper only uses the three datasets, and it cannot represent all medical image segmentation tasks.)

---

> ### Author Response · Authors · 2023-11-19
> **Official comment by authors**
>
> We thank reviewer sKkf to consider our work as well organized and clearly written. Here is our response regarding your comments on our limitations and additional result on a smaller dataset.
> > **The did not mention the limitation of their method**
>
> * We thank you for highlighting the importance of discussing the limitations of our work. We have now addressed this in our revised manuscript (Appendix A.4). We have identified two areas where we could conduct future studies on:
>   * 1). **Exploring scaling MGDC-UNet for building a foundation model in medical image segmentation**: The initial design of MGDC-UNet incorporated transformer components to improve feature learning. Recognizing the potential benefits of parameter scaling, our future research will focus on scaling up MGDC-UNet using the EfficientNet's compound scaling technique, aiming to create a more robust and versatile foundational CNN model for medical image segmentation.
>   * 2). **A more dynamic kernel strategy**: We found that larger kernels could enhance performance by effectively enlarging the Effective Receptive Fields (ERFs) but also increasing computational complexity. To optimize this, we could utilize a heterogeneous kernel strategy, akin to that used in the Inception network. This approach will combine small and large kernels, enabling the effective processing of multi-scale information without significantly boosting computational demands.
>
> > **They did not add results of their model for small dataset like MRBrainS dataset and MICCAI iSEG dataset (minor issue).**
>
> * We thank you for raising this concern and agree that adding a small dataset can further establish our method’s superiority. We decided to conduct further experiments on Prostate158 [1], a relatively smaller dataset (158 patients) with two challenging tasks: **1). Prostate gland segmentation and 2). Clinically significant prostate cancer segmentation**. Accurately segmenting the prostate and prostate cancer from MRI is difficult due to unclear boundaries with adjacent tissues and heterogeneous textures of the cancerous regions. We have updated our manuscript to include results from this dataset (Appendix A.3). Our experiments still confirmed that MGDC-UNet maintains excellent performance, even on this smaller and more challenging benchmark. For prostate region segmentation, small kernel MGDC-UNet (k=3) significantly outperformed all comparable methods, achieving a DSC of 0.855 and HD95 of 5.06. Further increasing kernel size to 5 led to 1.0% percent improvement in DSC and best performance was achieved at k=7, with 0.866 DSC. For prostate cancer segmentation, the small kernel MGDC-UNet (k=3) outperformed other state-of-the-art methods, achieving a DSC of 0.515 and HD95 of 7.77. Increasing kernel size to 5 and 7 further led to 0.9% and 1.2% improvement in DSC. These results reinforce the capability of MGDC-UNet to capture complex pathological features beyond those present in the datasets initially used for evaluation.
>
> > **could you change your title? Because (This paper only uses the three datasets, and it cannot represent all medical image segmentation tasks.)**
>
> * Thank you for your valuable feedback regarding the title of our paper. We understand your concern about the title potentially suggesting a broader range of testing than what has been covered in the paper. We will consider revising our title to "MGDC-UNet: Advancing 3D Medical Image Segmentation with Multi-Group Deformable Convolution" for the camera ready version upon acceptance.
> That said, we must also note that due to recent changes in ICLR policy, we are unable to modify the title or abstract of our submission at this stage. We hope that the content and discussions within the paper adequately convey the scope and limitations of our study, despite the constraints on altering the title.
>
> * We would like to emphasize that our study has indeed explored MGDC-UNET across a variety of segmentation tasks, including multi-organ segmentation, brain tumor segmentation, cross-modality segmentation (and prostate segmentation with the supplementary dataset). Moreover, our method has been tested on diverse modalities such as brain multi-parametric MRI, abdominal CT, and abdominal MRI (further extended to prostate MRI). These varied applications demonstrate the versatility of our method within the medical imaging domain.
> We appreciate your insightful feedback, which has highlighted important aspects of our work. We hope that our response clarifies the intent and scope of our study.
>
> We appreciate your feedback, which has prompted a more in-depth reflection on our work and guided our future research directions. We believe these additions to our manuscript will provide a more comprehensive understanding of MGDC-UNet's capabilities and limitations.
>
> Reference: 1. Adams, L.C., et al., Prostate158-An expert-annotated 3T MRI dataset and algorithm for prostate cancer detection. Computers in Biology and Medicine, 2022. 148: p. 105817.

---

> > ### Comment · Reviewer_sKkf · 2023-11-20
> >
> > Thank you for responding to my review comments and those of other reviewers and for clarifying the ambiguities.. The rebuttal solve this issue in my comments. I will increase my original score

---

> > > ### Author Response · Authors · 2023-11-22
> > > **Thank you for your response**
> > >
> > > Thank you for your feedback and for acknowledging our response to your comments and those of other reviewers. We're pleased that our rebuttal clarified the ambiguities. We sincerely appreciate your time and effort in reviewing our work.

---

### Official Review · Reviewer_Ateq · 2023-11-09

**Soundness:** 2 fair
**Presentation:** 2 fair
**Contribution:** 2 fair
**Rating:** 5
**Confidence:** 2

**Summary:**

This paper introduces MGDC-UNet, a method for 3D volumetric medical image segmentation that combines multi-group deformable convolution with transformer components. The goal is to address limitations in capturing complex spatial and semantic structures inherent in existing methods. MGDC-UNet employs deformable convolution operators with learnable spatial offsets, enhancing attention on semantically important regions. The approach leverages the stable spatial distribution across subjects to improve semantic learning.

While MGDC-UNet demonstrates superior accuracy on three challenging segmentation tasks—brain tumor segmentation (BraTS21), CT multi-organ segmentation (FLARE21), and cross-modality MR/CT segmentation (AMOS22)—it does have some limitations.

The novelty of the multi-group deformable convolution might be somewhat constrained, as it's a widely used technique in segmentation. In Figure 1, the results suggest that the proposed method's performance is comparable to self-attention. Similarly, in Figure 3, the results indicate that MGDC-UNet may not significantly outperform previous methods.

**Strengths:**

MGDC-UNet demonstrates superior accuracy on three challenging segmentation tasks.

**Weaknesses:**

The novelty of the multi-group deformable convolution might be somewhat constrained, as it's a widely used technique in segmentation. In Figure 1, the results suggest that the proposed method's performance is comparable to self-attention. Similarly, in Figure 3, the results indicate that MGDC-UNet may not significantly outperform previous methods.

**Questions:**

See comments.

---

> ### Author Response · Authors · 2023-11-19
> **Official comment by authors**
>
> We thank reviewer Ateq to acknowledge that our MGDC-UNet demonstrated superior accuracy on three challenging segmentation tasks. Here are our responses to the weaknesses of our paper:
> > **The novelty of the multi-group deformable convolution might be somewhat constrained, as it's a widely used technique in segmentation.**
> * While the concept of group convolution is indeed established in ConvNets for segmentation tasks, we believe that the concept of multi-group deformable convolution is not widely used in deformable convolution (DCN) or for medical image segmentation. In the related works of our paper, we discussed previous DCNs for medical segmentation tasks, such as DUNet [1] and Obelisk-net [2]. DUNet incorporated plain 2D DCN with spatial offsets to dynamically alter the receptive field for more accurate vessel segmentation. For Obelisk-net, they proposed to use 3D sparse DCN to replace the standard DCN to improve computational efficiency.
>
> * We summarized the following key differences between our MGDC and previous DCN works:
>   * **Masking as modulation strategy**: In MGDC, masks are used as a modulation strategy to guide the deformable convolution. These masks, normalized using SoftMax, allow for adaptive weighting of different groups within the feature maps. This approach is akin to the concept of attention mechanisms in transformers, providing fine-grained control over which parts of the input should be emphasized during the convolution process.
>   * **Depth-wise convolution for efficiency**: Unlike standard DCN, MGDC addressed the overparameterized process of computing offsets and masks by using depth-wise convolution, significantly reducing parameter counts without compromising the network's ability to learn complex features.
>   * **Multi-group mechanisms**: Our multi-group mechanism, inspired by successful models like ResNeXt [3], divides feature dimensions into groups, which compute deformable masks and offsets independently, facilitating diverse and intricate feature learning.
> * Finally, the efficiency of our MGDC operator allowed us to integrate transformer components, MLP and LayerNorm, into our MGDC-UNet architecture to advance feature learning capabilities, as seen in our ablation studies. We believe that this careful coordination of our operator and architecture design not only optimizes computational efficiency but also significantly enhances the model's capacity for accurate and intricate segmentation in medical imaging.
>
> > **In Figure 1, the results suggest that the proposed method's performance is comparable to self-attention.**
> * Thank you for your observation regarding Figure 1 and its comparison of our approach and self-attention method. We have updated Figure 1 to provide a clearer visual representation of the advantages offered by MGDC. From the updated figure, the top row showed that MGDC effectively enlarges ERFs while adapting to the geometries of various organs due to the learnable offsets. In comparison, while self-attention methods and large kernel methods both improved ERFs from small kernel methods, they could not adequately capture the complex semantic relationships among correlated organs. In the second row, we observed that sufficient ERF coverage on correlated organs leads to more accurate and coherent segmentation outcomes. Notably, MGDC-UNet demonstrated best segmentation performance since its ERF is suitably expanded to encompass correlated organs. The white arrows also point to specific instances where our method has resulted in more precise segmentation.
>
> > **•	Similarly, in Figure 3, the results indicate that MGDC-UNet may not significantly outperform previous methods.**
> * We appreciate your valuable feedback on Figure 3. Upon revisiting Figure 3, we understand your perspective that the differences in performance between MGDC-UNet and other methods may not seem substantial at first glance. Based on your suggestion, we have carefully revised the figure to include more demonstrative cases where the MGDC-UNet achieves notably fewer false positives in tumor subregion segmentation. As depicted in the first and second rows, our MGDC-UNet effectively minimizes false positive NCR (red) and ET (yellow) regions when segmenting brain tumors compared to competing methods. To maintain a balanced and fair comparison, the third row showcased MGDC-UNet exhibited moderate differences in tumor segmentation.
> * Despite the visual similarities in certain cases, **our method has statistically outperformed all existing methods (p<0.05, Student's t-test)**. This statistical validation is crucial and supports our claim that MGDC-UNET is not only effective in achieving superior segmentation accuracy but also holds significant clinical impact.
>
> We thank you for your invaluable feedbacks to our paper and looking forward to your response!

---

> > ### Author Response · Authors · 2023-11-19
> > **Official comment by authors**
> >
> > Refences:
> >
> > 1.	Jin, Q., et al., DUNet: A deformable network for retinal vessel segmentation. Knowledge-Based Systems, 2019. 178: p. 149-162.
> >
> > 2.	Heinrich, M.P., O. Oktay, and N. Bouteldja, OBELISK-Net: Fewer layers to solve 3D multi-organ segmentation with sparse deformable convolutions. Medical image analysis, 2019. 54: p. 1-9.
> >
> > 3. Xie, S., et al. Aggregated residual transformations for deep neural networks. in Proceedings of the IEEE conference on computer vision and pattern recognition. 2017.

---

> > ### Comment · Reviewer_Ateq · 2023-11-22
> >
> > Appreciate the author's prompt response. Many of my concerns have been effectively addressed. However, I still have a few remaining questions:
> > Structural Choices in Baseline:
> > It's notable that the author exclusively incorporates the multi-group deformable convolution into the U-Net as the baseline. Considering the potential impact, could other robust baselines also benefit from this specific structural augmentation?
> >
> >
> > Computational Cost of Multi-group Deformable Convolution:
> > The discussion should include a consideration of the computational cost associated with the multi-group deformable convolution. Given the inherently resource-intensive nature of deformable convolution, it's crucial to address whether the multi-group deformable convolution might incur excessive computational overhead.
> >
> >
> > Acknowledgment of Contribution in Multi-Scale Deformable Convolution:
> > The multi-scale deformable convolution finds wide application in computer vision tasks, as evidenced by its usage in [1][2][3][4]. It would enhance the clarity of the paper if the author explicitly outlines how their work contributes to or extends the existing landscape in this regard.
> >
> >
> > Comparison with Transformer and Positional Encoding:
> > While the deformable convolution adapts sampling positions based on learnable offsets, it's intriguing to consider the transformer's approach, which introduces positional encoding. Can the transformer entirely replace the deformable convolution, or is there potential for the proposed technique to complement and enhance transformer-based networks? Clarification on this aspect would be beneficial.
> >
> > [1]A multi-scale deformable convolution network model for text recognition,
> > [2]Multi-Scale Deformable CNN for Answer Selection,
> > [3]A U-Net Based Multi-Scale Deformable Convolution Network for Seismic Random Noise Suppression,
> > [4]DefA-Net: A Multi-Scale Deformable Attention Based MRI Image Segmentation Network

---

> > > ### Author Response · Authors · 2023-11-22
> > > **Official comment by authors [1/2]**
> > >
> > > Thank you for your reply. Here we provide further clarifications on your questions:
> > >
> > > > **It's notable that the author exclusively incorporates the multi-group deformable convolution into the U-Net as the baseline. Considering the potential impact, could other robust baselines also benefit from this specific structural augmentation?**
> > > * Thank you for your insightful query regarding the exclusive use of U-Net in our study. We chose U-Net as our baseline due to its proven effectiveness in medical image segmentation, which aligns with our main application for this paper. However, we recognize the potential benefits of integrating MGDC with other architectures.
> > > In particular, architectures like PSPNet [5], known for its pyramid pooling module, could potentially benefit from MGDC's ability to adaptively focus on relevant features, enhancing its scene parsing capabilities. Similarly, Inception Net [6], with its multi-scale convolutional approach, could leverage MGDC to optimize network efficiency and accuracy by effectively handling varying feature sizes.
> > > * As mentioned in Appendix A.4, in future work, we aim to integrate MGDC with Inception Net to effectively utilize its different kernel sizes, which we believe will further optimize network efficiency and accuracy, particularly in handling complex image segmentation tasks with varied feature dimensions.
> > >
> > > > **Computational Cost of Multi-group Deformable Convolution: The discussion should include a consideration of the computational cost associated with the multi-group deformable convolution. Given the inherently resource-intensive nature of deformable convolution, it's crucial to address whether the multi-group deformable convolution might incur excessive computational overhead**
> > > * Thank you for your insightful comment regarding the computational cost associated with MGDC. First, we kindly refer you to the Table 4 of the paper which detailed the parameter counts in our ablation study. We found that MGDC-UNet still compared favorably with 3D DCN in terms of parameter counts and accuracy.
> > > * However, to fully address your concern, we have conducted a thorough analysis of the computational costs in terms of both parameter counts and FLOPs, focusing on the network encoder on the BRATS dataset. We chose network encoder since all our architectural improvements were based on the encoder.
> > > |Encoder|Params (M)| FLOPs (G)|
> > > |--|--|--|
> > > |3D DCN|19.77| 11.3|
> > > |MGDC, without MLP and Layernorm|6.13|5.23|
> > > |MGDC, with MLP and Layernorm|9.27|7.51|
> > > * Our analysis reveals that MGDC significantly reduced parameter counts and computational complexities as compared to traditional DCN. Even after incorporating transformer components, the parameters and FLOPs are still 53.1% and 33.5% lower than the 3D DCN encoder. This demonstrates that MGDC thus strikes a balance between computational efficiency and the advanced feature learning capabilities. This balance is crucial for practical deployment in medical imaging scenarios, especially where computational resources are a limiting factor.
> > >
> > > > **Acknowledgment of Contribution in Multi-Scale Deformable Convolution: The multi-scale deformable convolution finds wide application in computer vision tasks, as evidenced by its usage in [1][2][3][4]. It would enhance the clarity of the paper if the author explicitly outlines how their work contributes to or extends the existing landscape in this regard.**
> > > * Thank you for your valuable feedback. The referenced papers highlight the versatility of multi-scale deformable convolution across various domains, from text recognition to seismic data processing and medical image segmentation. While MGDC also adopted a multi-scale DCN approach, our method represents a significant advancement in the following aspects:
> > >   * 1). improving DCN efficiency with depth-wise convolution,
> > >   * 2). adding multi-group mechanism facilitating diverse feature learning, and
> > >   * 3). integrating transformer components for advanced modeling capabilities.
> > > * While the referenced papers mentioned similar DCN works in various domains, our work concentrates on improving this method in the context of medical imaging for recognition of intricate spatial structures for precise segmentation. Our results show that MGDC-UNet achieves superior performance with significantly lower computational costs compared to traditional models, contributing a novel approach that balances efficiency and accuracy in medical image analysis.
> > > * We acknowledge your point regarding the clarity of our contribution within the existing landscape. Our paper aims to extend the application of multi-scale deformable convolution to a domain where precision and efficiency are critical. By integrating MGDC in a U-Net architecture, we have developed a solution that not only addresses the specific challenges of medical image segmentation but also demonstrates the potential to be deployed in clinical settings with limited resources.

---

> > > > ### Author Response · Authors · 2023-11-22
> > > > **Official comment by authors [2/2]**
> > > >
> > > > > **Comparison with Transformer and Positional Encoding: While the deformable convolution adapts sampling positions based on learnable offsets, it's intriguing to consider the transformer's approach, which introduces positional encoding. Can the transformer entirely replace the deformable convolution, or is there potential for the proposed technique to complement and enhance transformer-based networks? Clarification on this aspect would be beneficial.**
> > > > * Thank you for your query regarding the distinctions between deformable convolution and transformer positional encoding. First, we highlight their main differences:
> > > >   * DCN adapts sampling locations based on learnable offsets, which can be crucial for localizing and focusing on irregular, complex structures in medical images. This adaptability is particularly beneficial for segmenting images with varied and non-uniform features, commonly found in deformed organs or tumor’s heterogeneous textures.
> > > >   * Transformers, with positional encoding, excel in capturing global contextual information. While this also allows recognition of complex structures through larger receptive fields, we note that transformers are also plagued by the quadratic computational complexity with higher resolution images. More importantly, transformers lack the spatial inductive bias in convolution, which may require more training data to learn patterns that convolution methods infer more naturally.
> > > > * Therefore, we posit that transformers cannot entirely replace deformable convolutions in tasks like medical image segmentation. While transformers provide a broad understanding of the image, deformable convolutions offer precise, adaptive focus on critical areas.
> > > > * However, we acknowledge the distinct strengths from DCN and transformers. We agree that integrating these two approaches can yield a network that leverages both local adaptability and global information. Our future work could include a study to combine ViTs and MGDCs effectively, drawing inspirations from CMT [6].
> > > >
> > > > References:
> > > > 1. A multi-scale deformable convolution network model for text recognition.
> > > > 2. Multi-Scale Deformable CNN for Answer Selection.
> > > > 3. U-Net Based Multi-Scale Deformable Convolution Network for Seismic Random Noise Suppression.
> > > > 4. DefA-Net: A Multi-Scale Deformable Attention Based MRI Image Segmentation Network.
> > > > 5. Pyramid Scene Parsing Network. Hengshuang Zhao, Jianping Shi, Xiaojuan Qi, Xiaogang Wang, Jiaya Jia; Proceedings of the IEEE Conference on Computer Vision and Pattern Recognition (CVPR), 2017, pp. 2881-2890
> > > > 6. Guo, Jianyuan, et al. "Cmt: Convolutional neural networks meet vision transformers." Proceedings of the IEEE/CVF Conference on Computer Vision and Pattern Recognition. 2022.

---

### Official Review · Reviewer_spqz · 2023-11-09

**Soundness:** 3 good
**Presentation:** 3 good
**Contribution:** 2 fair
**Rating:** 5
**Confidence:** 4

**Summary:**

In this paper, the authors introduced MGDC-UNet, a multi-group deformable convolution network for 3D volumetric medical image segmentation. MGDCUNet employs deformable convolution operators with learnable spatial offsets to improve attention on semantically important regions. Meanwhile, they leverage stable spatial distribution across subjects to enhance semantic learning. In addition, they also incorporate transformer components to augment feature learning and reduce inductive biases inherent in traditional CNNs.

**Strengths:**

The network can adaptively adjust the offset of sampled locations, concentrating its attention on semantically relevant organ positions. Furthermore, the authors dynamically adjust offsets and modulation scalars to mitigate the inductive biases inherent in traditional CNNs, achieving transformer-like spatial aggregation. Also introduced MGDC blocks with a hybrid deformable convolution and multi-layer perceptron (MLP) structure for effective channel scaling and enhanced feature learning.

**Weaknesses:**

1. The originality of the algorithm's contribution remains ambiguous. Both deformable convolution and MLP are well-established application techniques, and while the authors have successfully combined them to achieve improved performance, their original contribution appears relatively modest.
2. More experimental details should be provided to facilitate replication of the proposed method as well as a fair assessment of the performance of all comparison methods, e.g. batch size, learning rate, loss function, etc.
3. The authors should have given more details about the data strategy, how the training dataset, validation set, and test set were divided, and how the optimal model was selected for performance comparison.
4. The datasets utilized in the experiments of the paper all feature objects with clear boundaries that are easy to segment. Given the authors' claim that their method can better capture complex pathological features, for a more comprehensive evaluation of the algorithm's performance, it would be advisable for the authors to attempt performance assessment on datasets with complex pathological and morphological characteristics, such as retinal OCT image segmentation, for example the dataset of RETOUCH -The Retinal OCT Fluid Detection and Segmentation Benchmark and Challenge.

**Questions:**

1. The originality of the algorithm's contribution remains ambiguous. Both deformable convolution and MLP are well-established application techniques, and while the authors have successfully combined them to achieve improved performance, their original contribution appears relatively modest.
2. More experimental details should be provided to facilitate replication of the proposed method as well as a fair assessment of the performance of all comparison methods, e.g. batch size, learning rate, loss function, etc.
3. The authors should have given more details about the data strategy, how the training dataset, validation set, and test set were divided, and how the optimal model was selected for performance comparison.
4. The datasets utilized in the experiments of the paper all feature objects with clear boundaries that are easy to segment. Given the authors' claim that their method can better capture complex pathological features, for a more comprehensive evaluation of the algorithm's performance, it would be advisable for the authors to attempt performance assessment on datasets with complex pathological and morphological characteristics, such as retinal OCT image segmentation, for example the dataset of RETOUCH -The Retinal OCT Fluid Detection and Segmentation Benchmark and Challenge.

---

> ### Author Response · Authors · 2023-11-19
> **Official comment by authors [1/2]**
>
> We thank reviewer spqz for the detailed review and constructive feedback on our submission. Below, we address your concerns and clarify the aspects you have highlighted.
>
> > **The originality of the algorithm's contribution remains ambiguous. Both deformable convolution and MLP are well-established application techniques, and while the authors have successfully combined them to achieve improved performance, their original contribution appears relatively modest.**
>
> * Thank you for raising concerns about the originality of our MGDC-UNET algorithm. We first highlighted our novelties as below:
>   * **Innovative MGDC operator**: Our primary contribution lies in the unique combination of deformable convolution with a multi-group mechanism, distinct from traditional deformable convolution approaches. Unlike standard DCN, MGDC addressed the overparameterized process of computing offsets and masks by using depth-wise convolution, significantly reducing parameter counts without compromising the network's ability to learn complex features. Furthermore, our multi-group mechanism, inspired by successful models like ResNeXt [1], divides feature dimensions into groups, which compute deformable masks and offsets independently, facilitating diverse and intricate feature learning.
>   * **Transformer Components for Advanced Feature Learning**: The efficiency of our MGDC operator allows us to integrate more complex components, MLP and LayerNorm, into our MGDC-UNet architecture to advance feature learning capabilities. We found that this integration did not introduce excessive parameter burdens, while significantly boosting the model's performance as seen in our ablation studies. We believe that this careful coordination of our operator and architecture design not only optimizes computational efficiency but also significantly enhances the model's capacity for accurate and intricate segmentation in medical imaging.
>   * **Insight into Kernel Size Exploration**: Another aspect of our contribution is the exploration of different kernel sizes to optimize the Effective Receptive Field (ERF). This approach is particularly relevant in medical imaging, where capturing intricate spatial relationships is paramount (as seen in Figure 1). Our experiments with various kernel sizes demonstrate that enlarging the ERF can indeed enhance the model's segmentation performance, a finding critical to the advancement of medical image segmentation techniques.
>
> * In summary, our MGDC-UNET combines an innovative MGDC operator to reduce parameters and facilitate complex feature learning by integrating transformer elements. Additionally, the exploration of various kernel sizes optimizes the ERFs, crucial for capturing detailed spatial relationships in medical imaging, thus significantly improving segmentation accuracy. Our work represents a significant step forward in the field, offering a novel and effective solution for complex segmentation tasks.
>
> > **More experimental details should be provided to facilitate replication of the proposed method as well as a fair assessment of the performance of all comparison methods, e.g. batch size, learning rate, loss function, etc.**
> * Thank you for your insightful comment regarding the need for more detailed experimental information in our paper.
> We would like to clarify that our training recipes including batch size, learning rate, loss function, and other relevant parameters, were indeed provided in our first submission (Table 5 of the Appendix). However, we recognize that this information might not have been immediately apparent or easily accessible to readers. To address this and improve the clarity of our manuscript, we have made the following modification:
>   * We updated the manuscript with more detailed explanation of hyperparameters used in training MGDC-UNet in **Appendix A.2**. We provided information such as batch size, loss function, network depths, channels and channel groups etc.
>   * We have included an explicit reference to **Appendix A.2.** in the main text of the paper, specifically in the section 4.1 discussing the implementation details.
>
> > **The authors should have given more details about the data strategy, how the training dataset, validation set, and test set were divided, and how the optimal model was selected for performance comparison.**
> * We thank you for your suggestion. In the original submission, we mentioned the use of a 5-fold cross-validation procedure to ensure the robustness and generalizability of our results. To further clarify this process, we have now included these information in **Appendix A.1** of our revised manuscript. This table details the exact distribution of the training, validation, and test sets for each dataset used in our study. For choosing the best model for evaluation, we used earlystopped the best model on validation performance. We also included vital information such as input image modality and evaluation metrics.

---

> > ### Author Response · Authors · 2023-11-19
> > **Official comment by authors [2/2]**
> >
> > > **The datasets utilized in the experiments of the paper all feature objects with clear boundaries that are easy to segment. Given the authors' claim that their method can better capture complex pathological features, for a more comprehensive evaluation of the algorithm's performance, it would be advisable for the authors to attempt performance assessment on datasets with complex pathological and morphological characteristics, such as retinal OCT image segmentation, for example the dataset of RETOUCH -The Retinal OCT Fluid Detection and Segmentation Benchmark and Challenge.**
> >
> > * Thank you for your your suggestion to evaluate our MGDC-UNet on datasets with more complex pathological and morphological characteristics, such as the RETOUCH. We recognize the importance of testing segmentation algorithms on a variety of datasets to ensure robustness and versatility.
> >
> > * Firstly, we would like to point out that our submission already includes comprehensive experiments on three large datasets (BraTS, FLARE, AMOS) that are well-recognized benchmarks for 3D medical image segmentation. These datasets have been utilized extensively in the literature, including in works such as UXNET [2], MedNEXT [3] and SwinUNETR [4] to evaluate the performance of segmentation methods. While the boundaries in these datasets may appear clear, the task of segmenting medical images accurately is nonetheless challenging and essential for clinical impact.
> > * We have made efforts to include additional datasets in our assessment. While we attempted to access the RETOUCH challenge dataset to extend our evaluation to retinal OCT images, unfortunately, we did not receive a response to our request, possibly due to the challenge already having concluded. This has limited our ability to perform the recommended analysis.
> >
> > * Nonetheless, to address your concern, **we have conducted additional experiments on the Prostate158 [5] dataset with the task of segmenting prostate gland and clinically significant prostate cancer.** Accurately segmenting the prostate and prostate cancer from MRI is difficult due to unclear boundaries with adjacent tissues and heterogeneous textures of the cancerous regions [6].
> >
> > * We have updated the manuscript to include results from Prostate158 (**Appendix A.3**). Our experiments still confirmed that MGDC-UNet maintains excellent performance, even on this smaller and more challenging benchmark.
> > * For prostate region segmentation, small kernel MGDC-UNet (k=3) significantly outperformed all comparable methods, achieving a DSC of 0.855 and HD95 of 5.06. Further increasing kernel size to 5 led to 1.0% percent improvement in DSC and best performance was achieved at k=7, with 0.866 DSC.
> > * For prostate cancer segmentation, the small kernel MGDC-UNet (k=3) outperformed other state-of-the-art methods, achieving a DSC of 0.515 and HD95 of 7.77. Increasing kernel size to 5 and 7 further led to 0.9% and 1.2% improvement in DSC. We believed that larger MGDC kernels capture a broader spatial context, which is beneficial for difficult segmentation tasks involving irregular and heterogeneous objects such as prostate gland and clinically significant prostate cancer. These results reinforce the capability of MGDC-UNet to capture complex pathological features beyond those present in the datasets initially used for evaluation.
> >
> > * In closing, we appreciate the opportunity to further substantiate the performance of MGDC-UNet and thank you for the suggestion to explore additional datasets. We believe that our method's strong performance across multiple benchmarks, including those with complex pathological features, attests to its robustness and effectiveness in medical image segmentation.
> >
> > References:
> >
> > 1. Xie, S., et al. Aggregated residual transformations for deep neural networks. in Proceedings of the IEEE conference on computer vision and pattern recognition. 2017.
> >
> > 2. Lee, H.H., et al., 3d ux-net: A large kernel volumetric convnet modernizing hierarchical transformer for medical image segmentation. arXiv preprint arXiv:2209.15076, 2022.
> >
> > 3. Roy, S., et al. Mednext: transformer-driven scaling of convnets for medical image segmentation. in International Conference on Medical Image Computing and Computer-Assisted Intervention. 2023. Springer.
> >
> > 4. Hatamizadeh, A., et al. Swin unetr: Swin transformers for semantic segmentation of brain tumors in mri images. in Brainlesion: Glioma, Multiple Sclerosis, Stroke and Traumatic Brain Injuries: 7th International Workshop, BrainLes 2021, Held in Conjunction with MICCAI 2021, Virtual Event, September 27, 2021, Revised Selected Papers, Part I. 2022. Springer.
> >
> > 5. Adams, L.C., et al., Prostate158-An expert-annotated 3T MRI dataset and algorithm for prostate cancer detection. Computers in Biology and Medicine, 2022. 148: p. 105817.
> >
> > 6. Zhu, Q., B. Du, and P. Yan, Boundary-weighted domain adaptive neural network for prostate MR image segmentation. IEEE transactions on medical imaging, 2019. 39(3): p. 753-763.

---

> > > ### Comment · Reviewer_spqz · 2023-11-22
> > >
> > > Thank authors for your comprehensive response. I have reviewed your detailed reply. My concerns have been solved. Consequently, I will increase my score.

---

> > > > ### Author Response · Authors · 2023-11-22
> > > > **Thank you for your response**
> > > >
> > > > Thank you for your reply. We greatly appreciate your thorough review and constructive feedback, which have undoubtedly enhanced the quality of our work. We are eagerly awaiting the updated review score. Thank you!

---

### Official Review · Reviewer_kXxV · 2023-11-16

**Soundness:** 2 fair
**Presentation:** 2 fair
**Contribution:** 2 fair
**Rating:** 6
**Confidence:** 4

**Summary:**

MGCD-UNet is introduced in this paper as a multi-group deformable convolution network for 3D volumetric image segmentation. The network modifies UNet by using deformable convolution operators, learnable spatial offsets, and a transformer-like architecture. The aim of this work is to address the limitations in semantic learning, especially with long-term dependencies, attention to important areas, inductive biases, stability, and complexity. The method was tested on three segmentation datasets where it demonstrated superior performance.

**Strengths:**

The paper addresses a challenging problem in medical image segmentation, and it demonstrates improvement in the three tested datasets in terms of accuracy.

**Weaknesses:**

The novelty of the paper is very limited as the method is a combination of many widely used techniques. Specifically, on page 5, the authors claim to develop a new architecture that includes a reverse bottleneck; however, this design is just similar to the inverted residual block used in MobileNetV2 [1]. Also, the authors mention on page 2 that they have developed a deformable convolution approach for 3D volumetric images, however, 3D deformable convolution has been applied before for videos in [2]. Moreover, the authors claim on page 3 that their method is improving computational efficiency, however, the only component that seems to decrease complexity is the depth-wise convolution. Following that, many components are added such as linear layers, SoftMax, transformer components, and an extra loop over “groups” which adds to the complexity of the network.  Tables 1 and 2 suggest that there is not much improvement in time and memory usage as other methods are comparable in performance (SegResNet inference time is 0.78 with memory usage of 3.3G while MGDC-UNET has inference time of 2.08 with 9.4G memory).
Time and memory usage were not mentioned in Table 3 which is inconsistent with the previous tables.
In Figure 1, we can clearly see that self-attention is comparable to the proposed MGDC. Figure 3 suggests a very small improvement but not significant. Similarly, in Figures 4 and 5, the improvement does not seem significant.
On page 6, the authors claim that increasing kernel size from 3 to 7 causes enhancements, however, some results in Table 2 disprove this claim.
The title is too general and gives a sense that the method is tested on many image segmentation tasks to prove its applicability, however, the method is only tested on 3 datasets, and it is not generalizable.

References:
[1] Mark Sandler, Andrew Howard, Menglong Zhu, Andrey Zhmoginov, and Liang-Chieh Chen.
Mobilenetv2: Inverted residuals and linear bottlenecks. In Proceedings of the IEEE conference
on computer vision and pattern recognition, pages 4510–4520, 2018.

[2] Xinyi Ying, Longguang Wang, Yingqian Wang, Weidong Sheng, Wei An, and Yulan Guo.
Deformable 3d convolution for video super-resolution. IEEE Signal Processing Letters, 27:1500–
1504, 2020.4

**Questions:**

Please refer to the above concerns.

---

> ### Author Response · Authors · 2023-11-19
> **Official comment by authors [1/4]**
>
> We thank reviewer kXxV for your valuable feedback. Below, we address your concerns and queries.
> > **The novelty of the paper is very limited as the method is a combination of many widely used techniques. Specifically, on page 5, the authors claim to develop a new architecture that includes a reverse bottleneck; however, this design is just similar to the inverted residual block used in MobileNetV2 [1].**
> * While we acknowledge the similarity of our MGDC block's inverted bottleneck design to the ones in MobileNetV2, we wish to highlight two novel aspects that distinguishes MGDC block from MobileNetV2 block:
>   * **1). Use of transformer components**: Our architecture’s uniqueness stems not just from the inverted bottleneck design but also from **the incorporation of MLP networks and LayerNorm**—elements not used in MobileNetV2. These components, inspired by ViTs, play a pivotal role in enhancing the model's performance, validated in our ablation studies (+ 0.5% DSC in BraTS and +0.8% DSC in FLARE21).
>   * **2). Use of multi-group deformable convolution**: Our primary contribution lies in the unique combination of deformable convolution with a multi-group mechanism, distinct from traditional DCNs. MGDC addressed the overparameterized process of computing offsets and masks by using depth-wise convolution, significantly reducing parameter counts without compromising the network's capacity. With our novel MGDC, our network can effectively enlarge the effective receptive fields (ERFs) to handle the complex spatial relationships in 3D medical images (Figure 1).
> * To clarify this in our manuscript, we propose revising the original sentence in page 5 as follows:
> *We present the MGDC Block, a new architecture that includes a reverse bottleneck design similar to MobileNetV2, but augmented with transformer components. While traditional inverted bottleneck design utilized depthwise convolution, our MGDC block leverages two MLP layers for channel expansion and reduction and LayerNorm for normalization, a design further inspired by Vision Transformers.*
>
> > **Also, the authors mention on page 2 that they have developed a deformable convolution approach for 3D volumetric images, however, 3D deformable convolution has been applied before for videos in [2].**
>
> * While we acknowledge that both our method and Ying et al’s method (D3D) used 3D deformable convolution, we highlight two important distinctions between MGDC and D3D:
>   * **1). How offset was computed for deformable convolution kernels**: D3D was developed specifically for 3D temporal data, which composed of a series of 2D images at different time scales. Hence, Ying et al. designed D3D for *super resolution task and only perform kernel deformation in spatial dimension to incorporate the temporal prior (i.e., frames temporally closer to the reference frame are more important) and reduce computational cost* [2]. Given the input feature of size C×T×W×H, D3D first computes offsets in size of 2N×T×W×H. Note that, the number of channels of these offset features is set to 2N for 2D spatial deformations (i.e., deformed along height and width dimensions). Then, the learned offsets are used to guide the deformation in a 2D sampling grid ( i.e., W and H dimensions), which produce the output feature map of D3D.
>   * **In contrast, our MGDC operator is specifically designed for 3D volumetric medical images**. Hence, MGDC computes offsets for 3D spatial deformations ( i.e., depth, width and height dimensions) to also account for the depth dimension characteristic of 3D medical images. Given the input feature of size C×T×W×H, our MGDC will compute offsets in size of 3N×T×W×H to guide the deformation in a 3D sampling grid, which eventually produce the output features of MGDC.
>
>   * **2). MGDC improves traditional DCN using depthwise convolution and multi-group mechanisms to decrease parameter counts**: Existing DCN methods such as D3D are not parameter efficient in computing feature maps, which only used plain convolutions. MGDC addresses this by adopting depth-wise convolution to reduce parameter counts. Also, we further incorporated multi-group mechanisms (as seen in ResNeXt) into DCN when computing offsets and modulation scalars. This allows our network to learn diverse and complex features more effectively. As shown in Table 4, adding multi-group mechanisms slightly reduced parameters while improving segmentation performances on both datasets.
> * Finally, to fully address your concern regarding the effectiveness of MGDC vs D3D, we added the following experiments comparing the two operators on BRATS 21:
> |Operator|DSC|HD95|
> |--|--|--|
> | D3D  | 0.892 | 7.249|
> | MGDC (k=3)| 0.906| 4.816|
>
> * We observed that D3D did not outperform MGDC, which we attributed to the core differences in how each algorithm was designed for each task. In summary, MGDC-UNet is distinctively tailored for 3D volumetric medical images with 3D spatial deformations and improved parameter efficiency.

---

> ### Author Response · Authors · 2023-11-19
> **Official comment by authors [2/4]**
>
> > **Moreover, the authors claim on page 3 that their method is improving computational efficiency, however, the only component that seems to decrease complexity is the depth-wise convolution. Following that, many components are added such as linear layers, SoftMax, transformer components, and an extra loop over “groups” which adds to the complexity of the network.**
> * Thank you for your insightful observations regarding the computational efficiency of our network. Here, we summarize the role of each component in MGDC block:
>   * 1). **Depth-wise Convolution**: As you rightly noted, the incorporation of depth-wise convolution is a key factor in reducing the network's parameters when computing offsets. As shown in Table 4 (row 1 and 2), introducing a shared weight mechanism to MGDC **decreased 22% parameters (from 71.5 millions to 58.7 millions)**.
>   * 2). **Multi-group mechanism**: Motivated by ResNeXt, we utilized group convolution to further improve the efficiency of deformable convolution. Instead of an “extra loop over groups”, our MGDC splits the feature dimensions into G groups, each of which computes its own deformable masks and deformable offsets. Compared to plain convolution in which every output channel is connected to every input channel, group convolution reduces computational load since each output channel is only connected to a subset of input channels (those within its group). As shown in Table 4 (row 3), adding multi-group mechanisms **slightly reduced the parameters by 0.6M** while improving DSC on both datasets (0.5% on BraTS21 and 0.2% on FLARE21).
>   * 3). **Linear Layers**: Similar to prior works in DCNv2 [3], our MGDC also utilized linear layers to compute offsets and modulation scalars (masks). In our method section, we compared our MGDC to a 3D implementation of DCN which also utilized linear layers to compute offsets and masks (not shown in manuscript due to page limit). Therefore, we respectively point out that this can hardly be considered as adding additional complexities to our network. For reference, we also attached github for DCNv2 implementation: https://github.com/chengdazhi/Deformable-Convolution-V2-PyTorch/blob/7e69911acfca4183172b829c50b174af721310e8/modules/modulated_dcn.py#L144
>
>   * 4). **Softmax**: To stabilize training, SoftMax was used to normalize mask values so that they sum up to 1 across each group. This also allows adaptive weighting the importance of different groups of the feature maps, similar to multi-head self-attention. Since softmax function contains no learnable parameters, it doesn't contribute to the model's parameter count. Also, SoftMax only involves exponentiation and normalization of input values, which are not computationally intensive. Therefore, we believed that the inclusion of a SoftMax layer in a model does not add significant computational complexity.
>
>   * 5). **Transformer components**: While we acknowledge that incorporating transformer components indeed added extra parameters to our network, our motivation for adding such components is not to improve computational efficiency, **but to improve feature learning capabilities.** Motivated by ViT blocks, we integrated MLP layers and LayerNorm into our MGDC block to perform feature expansion and reduction (inverted bottleneck) design. As validated by our ablation studies, incorporating MLPs improved DSC by 0.5% in BRATS21 and by 0.8% in FLARE21. What’s more, we kindly note that even after introducing MLPs to MGDC, the parameter only increased to 61 M, which is still **15%** less than the original 3D DCN network.
>
> * To improve the clarity of the baseline 3D DCN method which we compared to, we modified the sentence in method 3.1 describing 3D DCN to: *Accordingly, we first take a 3D a dynamic deformable convolution network (3D DCN) (Zhu et al.
> (2019)) with adaptive sampling offsets and modulation masks to enhance the targeted segmentation
> tasks.*

---

> ### Author Response · Authors · 2023-11-19
> **Official comment by authors [3/4]**
>
> > **Tables 1 and 2 suggest that there is not much improvement in time and memory usage as other methods are comparable in performance (SegResNet inference time is 0.78 with memory usage of 3.3G while MGDC-UNET has inference time of 2.08 with 9.4G memory). Time and memory usage were not mentioned in Table 3 which is inconsistent with the previous tables.**
> * Thank you for your comments regarding the time and memory usage comparisons in Tables 1. While it is true that SegResNet is more efficient in these aspects, we note that MGDC-UNet **significantly outperforms SegResNet in terms of segmentation accuracy**, with an improvement of 1.1% in DSC and a reduction of 2.64 mm in HD95. In clinical applications, where segmentation accuracy is paramount, the superior performance of MGDC-UNet could be more beneficial, despite its higher computational requirements.
> * Furthermore, when considering the BRATS21 dataset, the best SOTA method is SwinUNETR. We note that MGDC-UNet not only surpasses SwinUNETR in segmentation accuracy but also demonstrates a 35% improvement in inference time and a 27% reduction in memory consumption. This comparison underscores MGDC-UNet's efficiency and effectiveness, particularly in a challenging dataset like BRATS21.
> * Regarding the absence of time/memory analysis for the FLARE21 and AMOS22 datasets in Table 2 and 3, this was primarily due to page limitations. However, to provide a comprehensive understanding, we have included the time and memory usage details for these datasets in **Appendix A.5 Table 8**. To provide a wholistic understanding of the benefits of our method, we have also included the performance gaps of each method to our MGDC-UNet (k=3). From Table 8, we observed that while SegResNet indeed demonstrated faster inference speed and less memory usage, its segmentation performance is not comparable to MGDC-UNet.
>
> > **In Figure 1, we can clearly see that self-attention is comparable to the proposed MGDC. Figure 3 suggests a very small improvement but not significant. Similarly, in Figures 4 and 5, the improvement does not seem significant.**
> * We thank you for bringing these to our attention. We have updated Figure 1 to provide a clearer visual representation of the advantages offered by multi-group deformable convolutions. From the top row, we observed that MGDC effectively enlarges ERFs while adapting to the geometries of various organs due to the learnable offsets. In comparison, while self-attention methods and large kernel methods both improved ERFs, they could not adequately capture the complex semantic relationships among correlated organs. In the second row, MGDC-UNet demonstrated best segmentation performance since its ERF is suitably expanded to encompass correlated organs.
> * For your concerns in Figure 3, 4 and 5, we also revised them include both cases with large improvements and cases with moderate improvement. Each row represents a distinct case, arranged in descending order of MGDC-UNET's relative improvement over other methods, with **the top row** depicting the most significant enhancement.
> * For figure 3 (first and second rows), our MGDC-UNet effectively minimizes false positive NCR (red) and ET (yellow) regions compared to competing methods. To maintain a balanced and fair comparison, the third row showcased MGDC-UNet exhibited moderate differences in tumor segmentation. For figure 4, the top row depicted MGDC-UNet effectively segmented pancreas (red) whereas other methods failed. For figure 5, the top row depicted MGDC-UNet effectively segmented stomach (green) whereas other methods failed.
>
> > **On page 6, the authors claim that increasing kernel size from 3 to 7 causes enhancements, however, some results in Table 2 disprove this claim.**
> * We acknowledge that in Table 2, increasing the kernel size did not uniformly improve DSC for the liver and pancreas. However, we found that **gradually enlarging the kernel size leads to increasing number of metrics being statistically significant against the best SOTA methods across all datasets**. For example, in Table 2, we found that enlarging kernel size from 3 to 5 significantly improved DSC on spleen. Further improving kernel from 5 to 7 significantly improved DSC on kidney, liver and SDC on spleen and on average.
> * Regarding your concern for Table 2, the absence of significant improvement for the liver at larger kernel sizes could be attributed to the network's performance nearing saturation, particularly as it was already achieving a high accuracy of 99.4%. We also found both average DSC and SDC showed slight increases when transitioning from smaller to larger kernels.
> * Our claim is substantiated by growing statistically significant metrics outperforming SOTA models and a general upward trend in average metrics. Despite some less notable improvements, the overall data confirms that larger kernel sizes notably enhance segmentation results in MGDC-UNET.

---

> > ### Author Response · Authors · 2023-11-19
> > **Official comment by authors [4/4]**
> >
> > > **The title is too general and gives a sense that the method is tested on many image segmentation tasks to prove its applicability, however, the method is only tested on 3 datasets, and it is not generalizable.**
> > * Thank you for your valuable feedback regarding the title of our paper. We understand your concern about the title potentially suggesting a broader range of topics than what has been covered. We will consider revising our title to "MGDC-UNet: Advancing 3D Medical Image Segmentation with Multi-Group Deformable Convolution" for the camera ready version upon acceptance. However, due to recent changes in ICLR policy, we are unable to modify the title or abstract of our submission at this stage. We hope that the content and discussions within the paper adequately convey the scope and limitations of our study, despite the constraints on altering the title.
> > * We would like to emphasize that our study has indeed explored MGDC-UNET across a variety of segmentation tasks, including multi-organ segmentation, brain tumor segmentation, cross-modality segmentation (and prostate segmentation with the supplementary dataset). Moreover, our method has been tested on diverse modalities such as brain multi-parametric MRI, abdominal CT, and abdominal MRI (further extended to prostate MRI with the supplementary dataset). These varied applications demonstrate the versatility of our method within the medical imaging domain.
> >
> > We appreciate your insightful feedback, which has highlighted important aspects of our work. We hope that our response clarifies the intent and scope of our study.
> >
> > References:
> >
> > 1. Mark Sandler, Andrew Howard, Menglong Zhu, Andrey Zhmoginov, and Liang-Chieh Chen. Mobilenetv2: Inverted residuals and linear bottlenecks. In Proceedings of the IEEE conference on computer vision and pattern recognition, pages 4510–4520, 2018.
> >
> > 2. Xinyi Ying, Longguang Wang, Yingqian Wang, Weidong Sheng, Wei An, and Yulan Guo. Deformable 3d convolution for video super-resolution. IEEE Signal Processing Letters, 27:1500– 1504, 2020.4
> >
> > 3. Zhu, X., et al. Deformable convnets v2: More deformable, better results. in Proceedings of the IEEE/CVF conference on computer vision and pattern recognition. 2019.

---

> ### Comment · Reviewer_kXxV · 2023-11-22
>
> Thank authors for your comprehensive response. I have reviewed your detailed reply. The provided response effectively addresses the concerns I previously raised, and the authors have clearly incorporated these clarifications into their paper. Consequently, I will be raising my score.

---

> > ### Author Response · Authors · 2023-11-22
> > **Thank you for your response**
> >
> > We greatly appreciate your acknowledgment of our rebuttal. Your initial feedback was crucial in guiding our revisions and ensuring the clarity of our paper. Thank you for your thoughtful and constructive review!

---

### Meta-Review · Area_Chair_P63A · 2023-12-06

**Metareview:**

This paper presents a method for medical image segmentation which combines multi-group with deformable convolution. The technical novelty of the method is limited. The authors use Unet as baseline which also makes it difficult to find out its value in other networks.
Moreover, is it reasonable for the authors to justify their method using the current datasets. Obviously, there are many different public medical image datasets.

**Justification For Why Not Higher Score:**

I would like to discuss on the work.  On the one side, the overall score of the paper is not low (6). However, I have some concerns the novelty and the validation.
In term of novelty, Reviewer kXxV has commented earlier on the limited novelty of the paper (Reivewer Ateq agreed), and the authors argue that they have two novel aspects: 1). Use of transformer components and 2). Use of multi-group deformable convolution. To me, the first point is not a strong point as many people know that combining transformer with CNN helps in many situations. So, the question is more on if such combination straight forward or something un-expected/novel. Reviewer Ateq is more negative on the second point as this has been used in multiple places. [1]A multi-scale deformable convolution network model for text recognition, [2]Multi-Scale Deformable CNN for Answer Selection, [3]A U-Net Based Multi-Scale Deformable Convolution Network for Seismic Random Noise Suppression, [4] DefA-Net: A Multi-Scale Deformable Attention Based MRI Image Segmentation Network. The authors seem to admit this in their response.

In term of validation, the authors use Unet as baseline. However, Unet is quite outdated and there are multiple new methods, such as Segform, etc. The reviewer also raised a question on why these datasets not other datasets.

Can I have your opinion on that?

**Justification For Why Not Lower Score:**

NA

---

### Decision · Program_Chairs · 2024-01-16

Reject